# Spermidine Supplementation Protects the Liver Endothelium from Liver Damage in Mice

**DOI:** 10.3390/nu13113700

**Published:** 2021-10-21

**Authors:** Genís Campreciós, Maria Ruart, Aina Anton, Nuria Suárez-Herrera, Carla Montironi, Celia Martínez, Natalia Jiménez, Erica Lafoz, Héctor García-Calderó, Marina Vilaseca, Marta Magaz, Mar Coll, Isabel Graupera, Scott L. Friedman, Joan Carles García-Pagán, Virginia Hernández-Gea

**Affiliations:** 1Barcelona Hepatic Hemodynamic Laboratory, Hospital Clínic, Health Care Provider of the European Reference Network on Rare Liver Disorders (ERN-Liver), 08036 Barcelona, Spain; gcamprecios@clinic.cat (G.C.); maria.ruart.millan@gmail.com (M.R.); anton@clinic.cat (A.A.); nurisuhe@gmail.com (N.S.-H.); lafoz@clinic.cat (E.L.); hgarcia@clinic.cat (H.G.-C.); mvilaseca7@gmail.com (M.V.); mmagaz@clinic.cat (M.M.); jcgarcia@clinic.cat (J.C.G.-P.); 2Institut d’Investigacions Biomèdiques August Pi i Sunyer (IDIBAPS), 08036 Barcelona, Spain; cmartinezs@clinic.cat (C.M.); mdcoll@clinic.cat (M.C.); igraupe@clinic.cat (I.G.); 3Centro de Investigación Biomédica Red de Enfermedades Hepáticas y Digestivas (CIBEREHD), 28029 Madrid, Spain; 4Pathology Department, Hospital Clínic, 08036 Barcelona, Spain; montironi@clinic.cat; 5Liver Cancer Translational Research Group, Liver Unit, IDIBAPS-Hospital Clínic, UB, 08036 Barcelona, Spain; 6Liver Unit, Hospital Clínic de Barcelona, 08036 Barcelona, Spain; njimeneze@clinic.cat; 7Medicine Department, Faculty of Medicine, University of Barcelona, 08036 Barcelona, Spain; 8Division of Liver Diseases, Icahn Medical School at Mount Sinai, New York, NY 10029, USA; scott.friedman@mssm.edu

**Keywords:** autophagy, mitophagy, LSEC, endothelial dysfunction, liver fibrosis

## Abstract

Chronic liver diseases are multifactorial and the need to develop effective therapies is high. Recent studies have shown the potential of ameliorating liver disease progression through protection of the liver endothelium. Polyamine spermidine (SPD) is a caloric restriction mimetic with autophagy-enhancing properties capable of prolonging lifespan and with a proven beneficial effect in cardiovascular disease in mice and humans. We evaluated the use of dietary supplementation with SPD in two models of liver disease (CCl_4_ and CDAAH diet). We analyzed the effect of SPD on endothelial dysfunction in vitro and in vivo. C57BL/6J mice were supplemented with SPD in the drinking water prior and concomitantly with CCl_4_ and CDAAH treatments. Endothelial autophagy deficient (Atg7endo) mice were also evaluated. Liver tissue was used to evaluate the impact of SPD prophylaxis on liver damage, endothelial dysfunction, oxidative stress, mitochondrial status, inflammation and liver fibrosis. SPD improved the endothelial response to oxidative injury in vitro and improved the liver endothelial phenotype and protected against liver injury in vivo. SPD reduced the overall liver oxidative stress and improved mitochondrial fitness. The absence of benefits in the Atg7endo mice suggests an autophagy-dependent effect of SPD. This study suggests SPD diet supplementation in early phases of disease protects the liver endothelium from oxidative stress and may be an attractive approach to modify the chronic liver disease course and halt fibrosis progression.

## 1. Introduction

Chronic liver injury is a highly prevalent disease that affects 1.5 billion people worldwide and is responsible for 2 million deaths/year [1]. The liver endothelium plays a significant role in coordinating the liver response to injury and initiating the fibrotic process [2,3,4]. In fact, hepatic endothelial dysfunction (ED), characterized by changes in the phenotype of liver sinusoidal endothelial cells (LSEC), is an early event that occurs following liver injury and precedes the activation of hepatic stellate cells (HSC) and the onset of liver fibrosis in many chronic liver diseases [2,3,5].

Therefore, early interventions intended to prevent ED at early phases of the disease may foster an integrated, homeostatic response to liver injury able to hamper liver fibrosis and hepatocyte damage [2,5,6,7,8].

Spermidine (SPD) is an evolutionary conserved polyamine shown to exert a protective and lifespan-extending effect in mammals [9]. It is considered a caloric restriction mimetic due to its substantial antiaging effects through anti-inflammatory and antioxidant properties, while also enhancing mitochondrial metabolic function and improving proteostasis and chaperone activity [10]. Low levels of SPD have been correlated with disease severity in patients with nonalcoholic fatty liver disease (NAFLD) and nonalcoholic steatohepatitis (NASH) [11] and SPD is reduced in patients with acute on chronic liver failure (ACLF) [12]. Interestingly, its bioavailability can be increased by dietary supplementation with enriched aliments such as soybeans, wheat germ, nuts and fermented products [13], and dietary intake of SPD has been inversely correlated with the risk of heart failure, acute coronary artery disease, stroke and death due to vascular diseases [14], suggesting a beneficial role of SPD in maintaining endothelial phenotype in different organs. However, no data regarding the specific effect of SPD on the liver endothelium have been reported.

In this study, we hypothesized that daily SPD supplementation may help preserve LSEC homeostasis and delay disease progression. Therefore, we have evaluated the possible use of SPD as a chemoprophylactic agent in vitro and in two different in vivo models of liver injury with varying levels of endothelial damage, and have specifically characterized its effects on ED.

## 2. Materials and Methods

### 2.1. Animals

The animals were housed in polycarbonate cages and maintained in a temperature- and light-controlled facility under standard food and water ad libitum. All the procedures were performed in accordance with Spanish legislation and approved by the Animal Research Committee of the University of Barcelona and were conducted in accordance with the European Community guidelines for the protection of animals used for experimental and other scientific purposes (EEC Directive 86/609). For in vivo experiments, the animals were randomly distributed between groups and the experimenters were blinded for liver analysis purposes. Between 3 and 12 animals per group were used in total tissue experiments. All the experiments were performed in male C57BL/6J mice 10–14 weeks old at the beginning of the experiments. Generation of endothelial cell-specific Atg7 knockout mice in the C57BL/6J background was performed as previously described [5].

CDAAH model: the mice received a choline-deficient, L-amino acid-defined, high-fat diet (CDAAH) (Brogaarden #A06071302; 60% kcal from fat) for 9 weeks as previously described [15]. The animals were sacrificed under ketamine/midazolam anesthesia.

CCl_4_ model: the mice received intraperitoneal injections of 10% CCl_4_ (Sigma) in an olive oil solution (0.5 μL CCl_4_/g body weight) every other day in the course of two weeks for a total of six injections [16]. The animals were sacrificed 48 h after the last dose under ketamine/midazolam anesthesia.

SPD supplementation in vivo: SPD (Sigma, #S0266-5G) was added to the drinking water at a final concentration of 3 mM. Water was replaced every 2–3 days and SPD was freshly added from a 1 M SPD aqueous stock solution (pH 7.4), which was kept at −20 °C for no longer than one month [17]. The 1 M SPD stock solution was prepared by adding 3.14 mL of SPD to a final volume of 20 mL of H_2_O, adjusting the pH to 7.4. The control mice received regular water.

### 2.2. Biochemical Measurements

Liver transaminases were analyzed in serum with the standard methods at the Hospital Clínic CORE lab. Blood glucose was measured at indicated intervals using an Accu-Chek compact glucometer (Roche Farma, Madrid, Spain).

### 2.3. Cell Lines and Culture Conditions

The chemicals were purchased from Sigma-Aldrich (St. Louis, MO, USA), except when otherwise specified. The culture media and supplements for the cell culture were from Gibco-Invitrogen (Carlsbad, CA, USA) and the plasticware was from TPP (Trasadingen, Switzerland). The TSEC murine liver sinusoidal endothelial cell line was kindly provided by Dr. Vijay H Shah [18] and cultured with the endothelial cell medium (ECM, ScienCell) supplemented with 5% fetal bovine serum (FBS), 1% penicillin/streptomycin (PS) and 1% endothelial cell growth supplement (ECGS). The human hepatocellular carcinoma immortal cell line with endothelial origin SK-HEP1 was cultured with DMEM/F-12 supplemented with 10% FBS and 1% PS [19]. The cells were maintained at 37 °C in a 5% CO_2_ atmosphere.

To evaluate the SPD’s potential protective effects on endothelial cells, the TSEC and SK-Hep1 (passages 4–7 in both cases) were seeded in 6-well plates in triplicates, grown for 24 h and pre-treated with 10 µM SPD for another 24 h. The media were then changed and H_2_O_2_ (Sigma) was added (400 µM for the TSEC and 300 µM for SK-Hep1) for another 24 h. The cells were then collected for further analysis by means of flow cytometry or Western blotting. For viability assays, the cells were cultured in 96-well plates in triplicates and treated as explained above. To block autophagy, some wells were also treated with lysosome inhibitor chloroquine (CQ, 20 µM) concomitantly with SPD. Viability was measured after 24 h of H_2_O_2_ addition using colorimetric 3-(4,5-dimethylthiazol-2yl)-5-(3-carboxymethoxyphenyl)-(4-sulfophenyl)-2H-tetrazolium assays (MTS; Promega, Madison, WI, USA) according to the manufacturer’s instructions. Optical densities were measured with a Biotek plate reader. For SPD effect on the autophagy flux, the cells were seeded in 6-well plates for 24 h and 10 µM SPD was added for another 24 h. Two hours before collection, the cells were treated with or without CQ (20 µM). The cells were then collected for Western blot analysis.

### 2.4. Flow Cytometry

TSEC were seeded in 6-well plates and treated with SPD and H_2_O_2_ as described above. Twenty-four hours after the addition of H_2_O_2_, the cells were washed and incubated with MitoSOXTM Red (2.5 μM, Life Technologies) for 10 min at 37 °C to measure the mitochondrial O_2_^–^ content. The cells were then washed, trypsinized (0.05 mg/mL trypsin-EDTA solution for 4′, Gibco), and centrifuged at 1000× *g* for 1 min. The cells were then re-suspended in PBS + 2% FBS with 1 μg/mL DAPI (4′,6-diamidino-2-phenylindole, Roche) and kept on ice in the dark. CCCP (carbonyl cyanide 3-chlorophenyl hydrazine, Sigma) was used as the positive control. The samples were acquired with a FACS Canto II flow cytometer (BD Bioscience) and the data were analyzed with the Flowjo 10.0 software, measuring the mean fluorescence intensity (MFI). All the changes were calculated relatively to the control group.

### 2.5. Western Blotting

The cells and the tissues were homogenized in a RIPA (radioimmunoprecipitation assay) buffer complemented with Halt Protease & Phosphatase inhibitors (ThermoFisher, Waltham, MA, USA) and the total protein was estimated using Bradford’s Reagent (Bio-Rad, Hercules, CA, USA) following the manufacturer’s instructions. Cell lysates were prepared in the Laemmli sample buffer (Bio-Rad) + β-mercaptoethanol, resolved by sodium dodecyl sulfate (SDS) polyacrylamide gel electrophoresis (PAGE), transferred onto polyvinylidene fluoride (PVDF) membranes and blocked with 5% bovine serum albumin (BSA) in TBS + 0.05% Tween 20. The primary antibodies were incubated overnight at 4 °C in the blocking buffer, washed in TBS + 0.05% Tween 20 and subsequently incubated with their appropriate secondary antibodies conjugated with horseradish peroxidase for 1 h at RT. After washing again with TBS + 0.05% Tween 20, the membranes were developed with the Immobilon^®^ Western chemiluminescent HRP substrate (Millipore, Burlington, MA, USA). Images were acquired with a LAS-4000 apparatus (Fujifilm, TDI, Alcobendas, Spain) and the measurements were performed with the Multi Gauge software from Fujifilm following the manufacturer’s instructions. Protein ratios were normalized to loading controls and fold change was calculated relative to the respective control group. The following primary antibodies were used for Western blotting: LC3B (Cell Signaling #2775, 1/1000, Danvers, MA, USA), SQSTM1/p62 (Cell Signaling #5114, 1/1000), anti-phospho-Ubiquitin (Ser65) (Millipore #ABS1513-I, 1/1000, Burlington, MA, USA), OXPHOS (Abcam #ab110413, 1/1000) and 3-Nitrotyrosine (Sigma #N5538, 1/1000). GAPDH (Santa Cruz #sc-32233, 1/1000), β-actin (Sigma #A2228, 1/1000), α-tubulin (Sigma #T9026, 1/1000), Ponceau (Sigma) and Coomassie stains (Bio-Rad) served as the loading controls.

To measure the autophagy flux, the cultured cells were treated with the lysosomal inhibitor CQ (20 µM) for 2 h before collection. CQ impedes degradation of the cargo in the autophagosomes leading to the accumulation of autophagic vacuoles, which are then measured by determining the amount of the lipidated form of the MAP1LC3B (LC3II) protein localized in the autophagic vacuoles membrane by Western blotting [20].

### 2.6. Histologic, Immunohistochemical and Immunofluorescence Studies

Liver samples were formalin-fixed, paraffin-embedded, and 4 µm sections were obtained. Histological analysis was performed on hematoxylin- and eosin- (H&E) stained slides by an expert pathologist blind to the treatments. The inflammatory infiltrates in the liver parenchyma were measured in the portal, pericentral and lobular area using a score from 0 to 3 according to the following criteria: 0 = none; 1 = mild; 2 = moderate; 3 = severe. The assessment of necrosis was performed as follows: 1 ≤ 5%; 2 = 5–29%; 3 ≥ 30% of the parenchyma. Ballooning of hepatocytes was considered as 1 ≤ 4%, mild; 2 = 5–15%, moderate; 3 ≥ 15%, marked. In the case of the CDAAH murine model, H&E stains were used to score for lobular inflammation (0–3), ballooning (0–2) and steatosis (0–3), for a total unweighted score ranging from 0 to 8.

Hepatic fibrosis was evaluated with 0.1% Sirius Red staining. Fifteen images at the 200× magnification were captured with a Zeiss Axiovert 135 microscope to quantify collagen fibers evaluated as the red stained area per total area with the Image J software.

For immunohistochemical studies, paraffin sections were deparaffinized in xylene, rehydrated through graded alcohols and submerged in water. Heat-induced antigen retrieval in 10 mM citrate buffer (pH 6) or 10 mM Tris-EDTA was performed in a steamer for 30 min. The slides were then washed and mounted into Sequenza Racks (FisherScientific, Hampton, NH, USA). Endogenous peroxidase activity was blocked by incubating sections in H_2_O containing 3% H_2_O_2_ for 15 min. Nonspecific binding was blocked by incubating sections for 30 min in a blocking solution and incubated overnight at 4 °C with the primary antibodies. The primary antibodies were visualized with a Dako Real Envision Detection System Peroxidase/DAB+ kit, and the slides were counterstained with hematoxylin, dehydrated and mounted in DPX (Sigma). The following primary antibodies were used for immunohistochemistry: von Willebrand Factor (vWF, Dako A0082, 1/100), CD32-B/FcγRIIb (Santa Cruz sc-13271, 1/500), αSMA (Dako M0851, 1/100) and Desmin (Dako M0760, 1/200). For both Sirius and the different immunohistochemistry stains, 10 images at 200× were acquired per sample with a Zeiss Axiovert 135 microscope and quantified with the Image J software.

For tissue immunofluorescence, we followed the protocol described above until the addition of the secondary antibodies, just skipping the peroxidase blocking step in the first day. The secondary antibodies were incubated for 45 min at RT in the dark. The slides were then washed, DAPI-stained, mounted with Fluoromount-G^®^ (SouthernBiotech, Birmingham, AL, USA) and kept at 4 °C in the dark until imaged. The following primary and secondary antibodies were used: Cytochrome c (BD Pharmingen #556432, 1/100, Franklin Lakes, NJ, USA), LAMP-1 (Santa Cruz sc-19992, 1/100, Santa Cruz, CA, USA), TOM20 (Santa Cruz sc-17764, 1/100) Alexafluor 488 Donkey anti-mouse (ThermoFisher A21202, 1/300) and Alexafluor 555 Donkey anti-rat (ThermoFisher A21434, 1/300).

### 2.7. TUNEL (Terminal Deoxynucleotidyl Transferase (TdT) dUTP Nick End Labeling)

TUNEL staining consists in the identification of DNA strand breaks by signaling free 3′-OH terminals with labeled nucleotides through an enzymatic reaction mediated by TdT. We used the In Situ Cell Death Detection Kit, POD (Sigma, St. Louis, MO, USA) following the manufacturer’s instructions. In brief, 4 µm paraffin sections were deparaffinized in xylene, rehydrated through graded alcohols and submerged in water. The liver sections were then incubated with 20 μg/mL proteinase K (Sigma-Aldrich) in a 10 mM Tris HCl buffer, pH 7.4, washed and incubated with 50 μL of the TUNEL reaction mix for 1 h at 37 °C in the dark. The TUNEL mix includes dUTP-FITC, TdT and appropriate buffers. After several washes with PBS, the slides were incubated with an anti-FITC antibody bound to horseradish peroxidase for 30 min at 37 °C. Finally, the sections were stained with 3,3′-diaminobenzidine, counterstained with hematoxylin, dehydrated in graded alcohols and mounted in DPX. Fifteen images were captured at 200× magnification and quantified with the Image J software.

### 2.8. Electron Microscopy

Scanning electron microscopy (SEM) was used to quantify liver sinusoidal fenestrae. The livers were perfused through the portal vein with a fixation solution containing 2.5% glutaraldehyde and 2% paraformaldehyde in 0.1 M sodium cacodylate, pH 7.4, and fixed overnight at 4 °C. The samples were washed three times with 0.1 M cacodylate buffer. The liver sections were then fixed with 1% osmium in the cacodylate buffer, dehydrated in ethanol and dried with hexamethyldisilazane. The blocks were mounted onto stubs and sputter-coated with gold. Ten images per animal were acquired at a resolution of 15,000× using a Jeol 6380 Scanning Electron Microscope (JEOL Ltd., Tokyo, Japan). The number and porosity (percentage of LSEC surface occupied by fenestrae) of liver sinusoidal fenestrations were quantified using the Image J software.

### 2.9. Measurement of the Mitochondrial Superoxide (O_2_^–^) Cellular Content in Liver Tissues

In situ mitochondrial superoxide (O_2_^−^) levels were evaluated in 10 µm fresh liver cryosections using the mitochondrial oxidative fluorescent probe MitoSOXTM Red Mitochondrial Superoxide Indicator (Life Technologies) as previously described [21]. In brief, the animals were sacrificed in pairs, one animal from each group (+/− SPD). The livers were quickly snap-frozen in dry ice and 10 µm cryosections were obtained, with a piece of both livers in each cryosection. The sections were then incubated for 10 min at 37 °C with a 5 µM MitoSOXTM solution and rapidly imaged. A total of six fields at 200× from five different slides (for a total of 30 images per animal) were randomly selected and fluorescent images were obtained with an Inverted Zeiss Axiovert Fluorescent microscope. Quantitative analysis was performed with the Image J software. In each experiment, the same threshold was set for all the images, and integrated density (the product of the area and the mean gray value) was measured. Each SPD-treated animal was compared to the vehicle-treated mouse sacrificed at the same time.

### 2.10. Quantitative Real-Time Polymerase Chain Reaction (qRT-PCR)

An RNeasy Mini kit (Qiagen) was used to isolate total RNA from tissues following the manufacturer’s instructions. First-strand cDNA was synthesized using a High Capacity RNA to cDNA conversion kit (Applied Biosystems) according to the manufacturer’s instructions. Quantitative real-time polymerase chain reaction (qRT-PCR) was performed with a PowerUp SYBR Green Master Mix (Applied Biosystems) with technical duplicates using an ABI Prism 7900 HT Cycler (Applied Biosystems). PCR cycle parameters were as follows: 95 °C for 10 min followed by 40 cycles at 95 °C for 15 s and 60 °C for 1 min. The SDS3.2 software (Applied Biosystems, Waltham, MA, USA) was used for analysis and the results calculated were based on the 2^−ΔΔCt^ method. Gapdh or Actb were used as the housekeeping genes. The results are shown as the fold change relative to the correspondent control group. Primer-specific sequences are shown in Appendix A.

### 2.11. Statistical Analysis

Statistical analysis was performed using GraphPad 5.01 for Windows. The groups were compared using Student’s *t*-test when comparing two groups where adequate. Fisher’s exact test was performed to evaluate the contingency tables when needed. All the data were reported as the means ± SEM. The differences were considered significant at a *p*-value ≤ 0.05.

## 3. Results

### 3.1. Prophylactic Treatment with SPD Improves the Endothelial Phenotype and the Response to Injury

We initially determined whether a prophylactic strategy (SPD pretreatment) had any protective effect on the LSEC phenotype and the response to injury. Because LSECs rapidly lose their phenotype when cultured in vitro [5,6] and there are no reliable markers to evaluate endothelial function in immortalized cell lines, we evaluated the direct effect of SPD pretreatment on two liver endothelial cell lines of murine (TSEC) and human (SK-HEP1) origin following an oxidative injury. We pretreated cells with SPD for 24 h and then mimicked the effect of high intracellular oxidative stress generation by directly treating each cell line with a concentration of H_2_O_2_, which led to ~50% cell death. As observed in Figure 1A, SPD pretreatment protected both endothelial cell lines significantly by improving their viability compared to the vehicle treatment. Concurrently, SPD pretreatment increased the autophagic flux in both cell lines, as measured by the accumulation of the LC3BII protein (Appendix A).

We next analyzed whether pretreatment with SPD in vivo prevented or ameliorated ED. The C57BL/6J mice were injected with CCl_4_ for 2 weeks (toxic model, low endothelial damage). SPD was given in the drinking water (3 mM) starting two weeks before the first CCl_4_ dose and during the duration of the study.

We analyzed LSEC dysfunction by measuring CD32b and glycoprotein vWF by means of immunohistochemistry, two well-established markers to evaluate endothelial differentiation and dysfunction [22,23]. SPD pretreatment resulted in increased levels of CD32b (Figure 1B), whereas vWF was slightly decreased (Figure 1C), both markers suggesting an improvement in the LSEC phenotype by SPD treatment in the CCl_4_ model. Supporting this notion, quantification of LSEC fenestrae by SEM, which is the “gold standard” to evaluate LSEC dedifferentiation [22,24], also displayed increased porosity and the number of fenestrae in SPD-treated livers (Figure 1D).

To assure that the observed changes were not model-specific, we decided to use a more chronic liver injury model with higher degree of endothelial damage and capillarization. We fed the C57BL/6J mice with a choline-deficient, L-amino acid-defined, high-fat diet (CDAAH) for 9 weeks (NASH model) [25] and added SPD in the drinking water starting two weeks before the addition of the CDAAH diet and during the 9 weeks of treatment. As observed in Figure 1 (panels E–G), addition of SPD also prevented the decrease of CD32b and the subsequent increase in vWF secretion in the liver endothelium, which also showed a higher degree of porosity and number of fenestrae when compared to livers from the mice not fed with SPD.

These results suggest that prophylactic SPD supplementation protects against liver endothelial injury both in vitro and in vivo.

### 3.2. Prophylactic SPD Ameliorates Liver Parenchymal Injury In Vivo

Besides improving ED, prophylactic treatment with SPD was accompanied by a significant decrease in collagen deposition in the CCl_4_ mice as determined by Sirius Red staining (Figure 2A) and minor changes in HSC activation as determined by αSMA staining, but no differences in Desmin (Figure 2B,C), suggesting an effect of SPD on HSC deactivation rather than on the number of HSC. SPD-treated animals also had a slight decrease in cellular apoptosis (TUNEL) and transaminases levels compared to their corresponding controls (Figure 2D,E).

In the CDAAH model, SPD was not able to decrease the fibrosis content (Figure 2F–H) despite its slight effect on HSC deactivation, but on the other hand, its effects on reducing cellular apoptosis were more remarkable (Figure 2I), with also a slight decrease in transaminases (Figure 2J). This result prompted us to take a closer look on other classical features associated with the development of NASH. Interestingly, in the CDAAH model, we observed that SPD treatment also reduced hepatocyte ballooning (Appendix A), one of the key histological findings of NASH that correlates with prognosis [26], and it alleviated the NASH-associated metabolic phenotype by reducing abdominal and epididymal fat content as well as by lowering blood glucose levels (Appendix A).

These results suggest that the improvement observed in both models by SPD prophylaxis may be broader than its effect on the liver endothelium.

### 3.3. The Beneficial Effect of SPD Is Partly Mediated by Endothelial Autophagy

Different studies have indicated that the majority of SPD effects in different systems are mediated by autophagy enhancement [10]. Indeed, in our in vitro models, SPD did increase autophagy levels of both cell lines and, in fact, its effects were dependent on autophagy, since addition of the autophagy inhibitor chloroquine concomitantly with H_2_O_2_ prevented SPD-mediated amelioration on either cell line (Figure 1A). Additionally, SPD had no effects in autophagy-deficient TSEC (Appendix A), reinforcing the concept that SPD protection in these cell lines is mediated by autophagy enhancement.

As stated above, LSEC are an integral part of the liver defense upon injury and we and others have recently shown that endothelial autophagy impairment leads to exacerbated liver injury in different models of liver disease [5,7]. Since we observed different responses to SPD in our two models of liver disease, coinciding with different degrees of ED, we decided to ascertain the specific importance of the improvement of the liver endothelium in SPD-treated mice in the CCl_4_ model. We used Atg7-flox/VE-Cadherin-Cre (Atg7endo) mice with a specific impairment of endothelial autophagy [5] and treated them with CCl_4_ for two weeks with or without SPD in the drinking water. Contrary to the observations in the wild-type mice, SPD treatment was unable to improve ED in the Atg7endo mice (Appendix A). SPD neither reduced cellular apoptosis (Appendix A) nor deactivated or decreased the number of HSC in Atg7endo livers (Appendix A), thereby preventing the improvement in the fibrosis degree observed in the wild-type mice (Appendix A and Figure 2L) and confirming the importance of the endothelium and of endothelial autophagy in particular in the maintenance of the liver homeostasis.

### 3.4. SPD’s Benefit Is Mainly Due to a Reduced Production of Oxidative Stress through Mitochondrial Protection

Oxidative stress is a major pathogenic event occurring during chronic liver injury regardless of the underlying etiology, and maintaining an adequate redox balance is essential to halting the onset and progression of liver fibrosis [27,28,29]. Because its antioxidant potential is a key property of SPD [30], we determined whether SPD pretreatment protected endothelial cells by decreasing oxidative stress in vitro. We collected cells 24 h after H_2_O_2_ addition and evaluated the levels of O_2_^–^ using flow cytometry (Figure 3A). As expected, the TSEC pretreated with SPD displayed lower levels of oxidative stress compared to those pretreated with H_2_O_2_ alone and similar to those observed in the vehicle-treated cells. SPD pretreatment induced a modest, nonsignificant increase in several classical and NRF2-mediated antioxidant enzymes (Appendix A), suggesting that SPD most probably acts through additional mechanisms besides augmenting, to a limited extend, the antioxidant capacity of endothelial cells.

SPD can react with reactive oxygen species (ROS) and act as an antioxidant directly [31]; however, the fact that in our system its effects depend at least in part on autophagy activation indicated an alternative mechanism; therefore, we focused on mitochondria. Besides being the major producers of ROS, mitochondria are also highly affected by ROS, and when dysfunctional, they are rapidly removed via selective autophagy (mitophagy) [32]. We first analyzed the amount of dysfunctional mitochondria by measuring the amount of phosphorylated ubiquitin chains at S65 (pS65-Ub) in our experimental system by means of Western blotting. Damaged mitochondria accumulate PINK1 in their surface, which phosphorylates the S65-Ub of PARKIN. PARKIN, in turn, phosphorylates other mitochondrial surface proteins at their S65-Ub, signaling for selective degradation of the mitochondria via mitophagy [33]. Addition of H_2_O_2_ to the TSEC significantly increased the amount of pS65-Ub and while pretreatment with SPD did not reach control levels, it tended to reduce the amount of dysfunctional mitochondria (Figure 3B). We explored whether this decrease in damaged mitochondria could reflect increased removal by mitophagy in SPD-pretreated cells. As observed in Figure 3C, however, SPD addition to the TSEC for 24 h did not lead to increased colocalization of TOM20 and LAMP-1, therefore indicating SPD might exert a direct protective effect on mitochondria by reducing intracellular oxidative stress through increased autophagy, rather than by inducing the removal of damaged mitochondria via mitophagy. Indeed, SPD increased the levels of mitochondrial DNA (Figure 3D) and the expression of mitochondrial complexes I and III of the mitochondrial oxidative phosphorylation system (OXPHOS) (Figure 3E), both of which reflect an overall increase in healthy mitochondria in the LSEC treated with SPD.

We then determined whether this mechanism was relevant in vivo in the CCl_4_ model of liver injury. Indeed, livers of the SPD-treated mice displayed reduced overall oxidative stress compared to the control group treated with CCl_4_ alone (Figure 4A), but this reduction was not mediated by increased expression of the classical or NRF2-mediated antioxidant enzymes as there were no differences between the SPD-treated and control livers (Appendix A).

This reduction of oxidative stress correlated with a slight decrease in dysfunctional mitochondria as the levels of pS65-Ub in total liver tissue were slightly improved upon SPD treatment (Figure 4B). In agreement, SPD treatment did not lead to increased removal of mitochondria by mitophagy (Appendix A); however, we observed increased levels of mitochondrial DNA (Figure 4C) and protein levels of complex I and III in those livers from the mice treated with SPD (Figure 4D), therefore recapitulating in the CCl_4_ murine model in vivo the results previously observed in vitro and suggesting an improvement in mitochondrial fitness upon SPD treatment.

In the CDAAH model, SPD produced effects similar to those described above in the CCl_4_ model, albeit to a lesser extent, with only minimal reductions in mitochondrial oxidative stress and dysfunction and improvements in the amount of mitochondrial DNA and protein levels of complexes I and III (Figure 4E–H and Appendix A).

To further confirm that the beneficial effect of SPD on decreasing oxidative stress and maintaining mitochondrial homeostasis was mediated at least in part by endothelial autophagy levels, we evaluated these parameters in endothelial autophagy-deficient mice. As shown in Appendix A, SPD treatment was unable to significantly reduce the oxidative stress levels produced by CCl_4_ toxicity in Atg7endo mice. SPD ameliorated the levels of pS65-Ub (Appendix A), indicating some reduction in the number of damaged mitochondria signaling for mitophagy. However, mitochondrial DNA and complex I and III levels remained unchanged (Appendix A, respectively), suggesting that endothelial autophagy is needed in the regulation of mitochondrial homeostasis. Although SPD may also confer an autophagy-independent effect, our data highlight the importance of endothelial autophagy in the antioxidant response and control of liver homeostasis and its response to injury.

Overall, these results point to the increase in autophagy and the subsequent reduction in intracellular ROS as the main mechanism by which SPD protects mitochondria, which remain more functional, attenuating the increase in ROS production and accumulation. This, in turn, reduces the stress in both the liver endothelium and the parenchyma.

## 4. Discussion

Chronic liver diseases and their complications are an important health burden worldwide in part due to the lack of effective treatments capable of reverting the complex derangements caused by the disease once these are fully established. Therefore, strategies aimed at preventing or delaying the onset and progression of the disease can be of great interest. In this context, LSEC stand out as an appealing target.

LSEC line the hepatic sinusoids and are the liver’s first barrier of defense [4,34,35]. We and others [5,7] have recently shown that endothelial autophagy is an essential pathway required for LSEC to maintain liver homeostasis early following a liver insult. Based on these studies, we hypothesized that potentiating endothelial autophagy during the initial stages of liver injury could have a beneficial effect on the endothelium and, consequently, on the whole liver. We evaluated the prophylactic treatment with the natural polyamine SPD, classified as a caloric restriction mimetic and a potent autophagy enhancer [10], and tested it in LSEC in vitro and in two different models of early chronic liver disease, the CCl_4_ toxic-induced model and the CDAAH-induced NASH model.

SPD treatment improved the liver endothelium in both animal models. Nevertheless, the improvement observed in each model differed substantially and depended on the overall degree of endothelial capillarization, which in turn correlated with the capacity of SPD to increase LSEC autophagy levels. These results are in agreement with studies showing autophagy as an important pathway against intracellular stress, particularly during the initial phases of endothelial injury, but not in advanced stages where its protective effects are overwhelmed by the accumulation of toxic products. Thus, as an autophagy enhancer, SPD might be effective in protecting the liver endothelium, especially in those cases where endothelial damage has not exceeded a determined threshold, and justifies why the effects are more pronounced in our CCl_4_ model (low endothelial damage) than in the CDAAH-fed mice (severe endothelial damage).

Recent findings have indicated that preserving endothelial phenotype at early phases of liver injury fosters an integrated, homeostatic response to liver damage and ameliorates fibrosis [2,5,7,27]. This is the case in the CCl_4_ model, where besides an improvement in the liver endothelium, prophylactic treatment with SPD reduces the activation of HSCs to ameliorate fibrosis. Whether this effect was obtained solely through SPD’s action on LSEC is unclear since SPD was administered systemically and could have direct effects on all cell types in the liver. However, the lack of effect of SPD on liver fibrosis in the CDAAH model with severe endothelial damage, but especially in Atg7endo mice where only endothelial autophagy is impaired, emphasizes the central role of LSEC as gatekeepers of liver homeostasis and the importance of preserving a functional endothelium to prevent fibrosis progression.

Concerns might be raised regarding the use of an autophagy enhancer in the liver since autophagy induces the activation of HSC [16,36]. However, our results indicate that the beneficial effects of increasing autophagy in the endothelium and other liver cell types [37,38] at early phases of the injury may hamper HSC activation.

Mice treated with SPD display improvement in the liver endothelial phenotype and also an improved parenchymal response to injury, mainly characterized by reduced apoptotic cell death and fibrosis. Our data suggest that most of SPD’s effects are due to a significant reduction of intracellular ROS, mediated primarily by reducing mitochondrial stress via autophagy activation and, to a lesser extent, by an SPD-mediated increase in antioxidant enzymes. Although not focused on the study of the endothelium, SPD has been tested in different models of liver disease and shown protection of the liver principally through the activation of autophagy and the antioxidant enzymes, including glutathione peroxidases and superoxide dismutases, but also through the NRF2-mediated antioxidant response [39]. In our animal models, the effect observed in the antioxidant response was not pronounced, possibly due to SPD’s prophylactic administration and the use of less severe models of fibrosis.

SPD also exerts cardiac and neuronal protection through activation of mitophagy and clearance of dysfunctional mitochondria [14,40]. Having healthy mitochondria is crucial for liver homeostasis. Mitochondrial dysfunction and reduction of mitochondrial OXPHOS complexes I and III are a hallmark of chronic liver diseases; it occurs early during disease onset and it is maintained through progression of the liver disease ([41] and unpublished data). We observed both reduced mitochondrial oxidative stress and the number of damaged mitochondria, although mitophagy was not increased in our animals. Therefore, we speculate that the early increase of macroautophagy in our model may have been responsible for direct scavenging of ROS, reducing their accumulation and preventing mitochondrial damage and dysfunction.

Finally, systemic administration of SPD improves the health of other tissues besides the liver, especially in the CDAAH model, where SPD decreased fat accumulation and lowered blood glucose levels, two important hallmarks of NASH pathogenesis. In the current context where many clinical trials are now seeking synergistic effects that may arise from combining two or several compounds on different pathways of liver disease simultaneously, dietary SPD supplementation in early phases of disease activity could improve endothelial function and attenuate liver fibrosis progression, especially in patients at risk of developing fibrosis or when detected at very early stages. SPD can be easily supplemented with the diet and has already been tested in humans with good tolerance and no side effects described [42,43].

In conclusion, in this study, we corroborate the importance of protecting the liver endothelium at early stages of liver injury and establish SPD as a potential candidate for prophylactic treatments, with the aim to prevent early endothelial dysfunction and liver damage progression.

## Figures and Tables

**Figure 1 nutrients-13-03700-f001:**
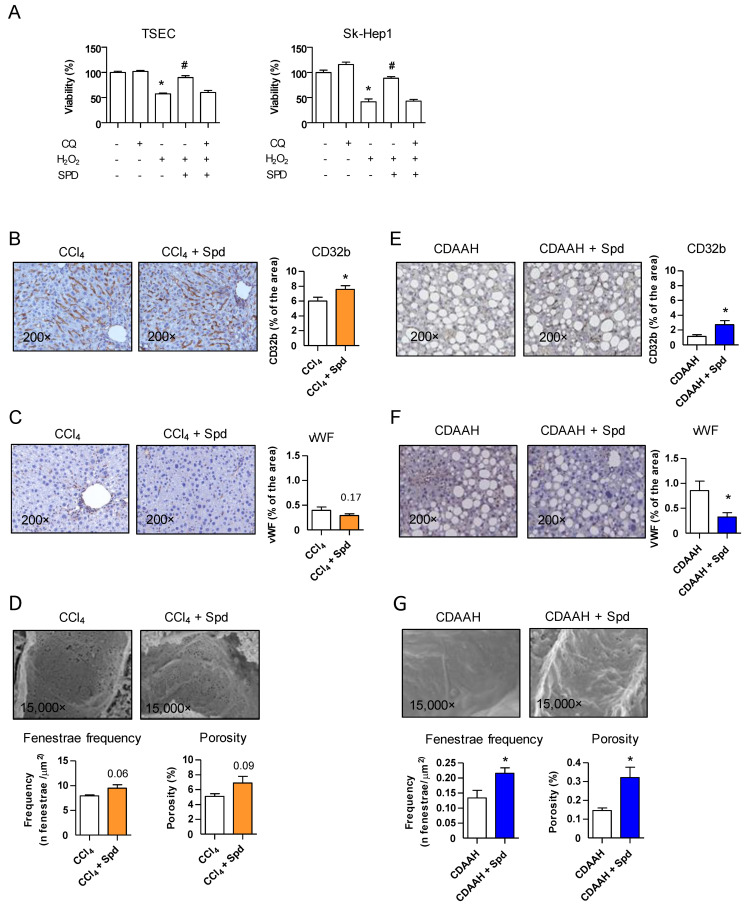
SPD improves the LSEC viability in vitro and the endothelial phenotype in vivo. (**A**) TSEC and SK-Hep1 viability studies upon H_2_O_2_ damage measured by MTS assay. The cells were seeded in 96-well plates in triplicates. The next day, the cells were pretreated with SPD or SPD + CQ for 24 h. The medium was then changed, and H_2_O_2_ was added for another 24 h before adding the MTS solution. One representative of the three independent experiments is shown. (**B**,**E**) CD32b immunohistochemistry and quantification in whole liver tissue sections from the CCl_4_ and CDAAH mice, respectively. (**C**,**F**) Immunohistochemistry and quantification of vWF in whole liver tissue sections from the CCl_4_ and CDAAH mice, respectively. (**D**,**G**) Fenestrae frequency and porosity were measured by SEM in livers from the CCl_4_ and CDAAH mice, respectively. For the IHC studies, *n* = 7–10 mice per group. For the SEM studies, *n* = 5–6 mice per group. The results are the means ± SEM; * *p* < 0.05 compared to the correspondent control; # *p* < 0.05 compared to the H_2_O_2_ treatment (Student’s *t*-test).

**Figure 2 nutrients-13-03700-f002:**
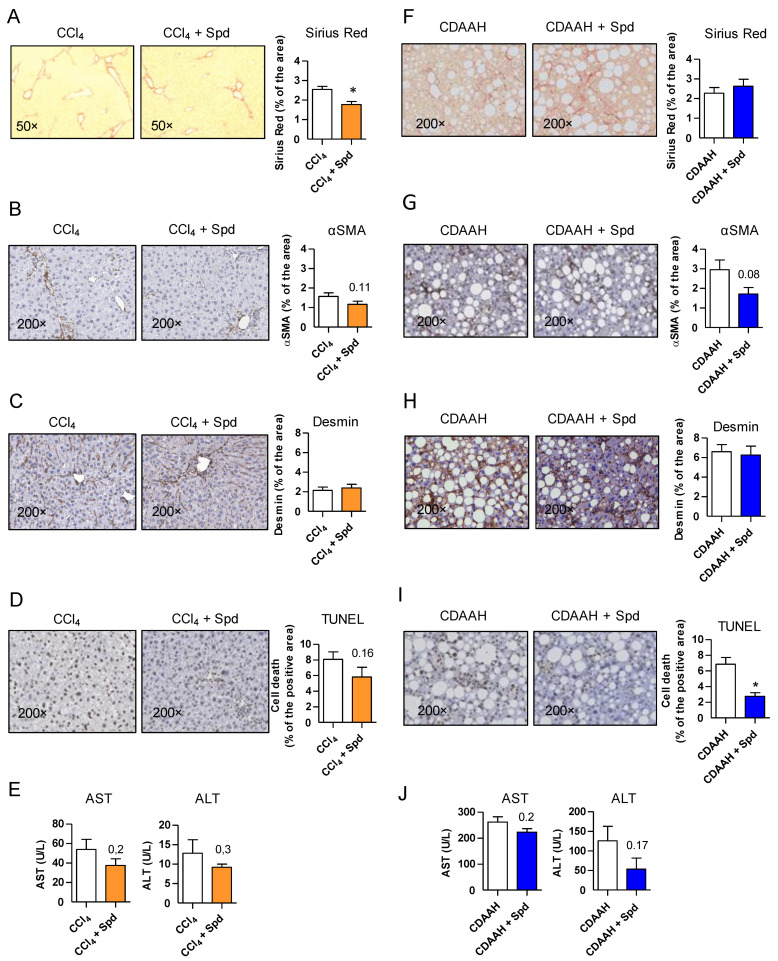
SPD improves the liver response to injury and attenuates fibrosis and HSC activation in vivo. (**A**,**F**) Sirius Red staining and quantification in whole liver tissue sections from the CCl_4_ and CDAAH mice, respectively. (**B**,**G**) Immunohistochemistry and quantification of αSMA in whole liver tissue sections from the CCl_4_ and CDAAH mice, respectively. (**C**,**H**) Desmin immunohistochemistry and quantification in whole liver tissue sections from the CCl_4_ and CDAAH mice, respectively. (**D**,**I**) TUNEL assay performed on liver tissue sections from the CCl_4_ or CDAAH mice, respectively. (**E,J**) Serum ALT and AST transaminases from the mice treated with the CCl_4_ and CDAAH diet with or without SPD supplementation. For each study, *n* = 7–12 mice per group. The results are the means ± SEM; * *p* < 0.05 compared to the vehicle (Student’s *t*-test).

**Figure 3 nutrients-13-03700-f003:**
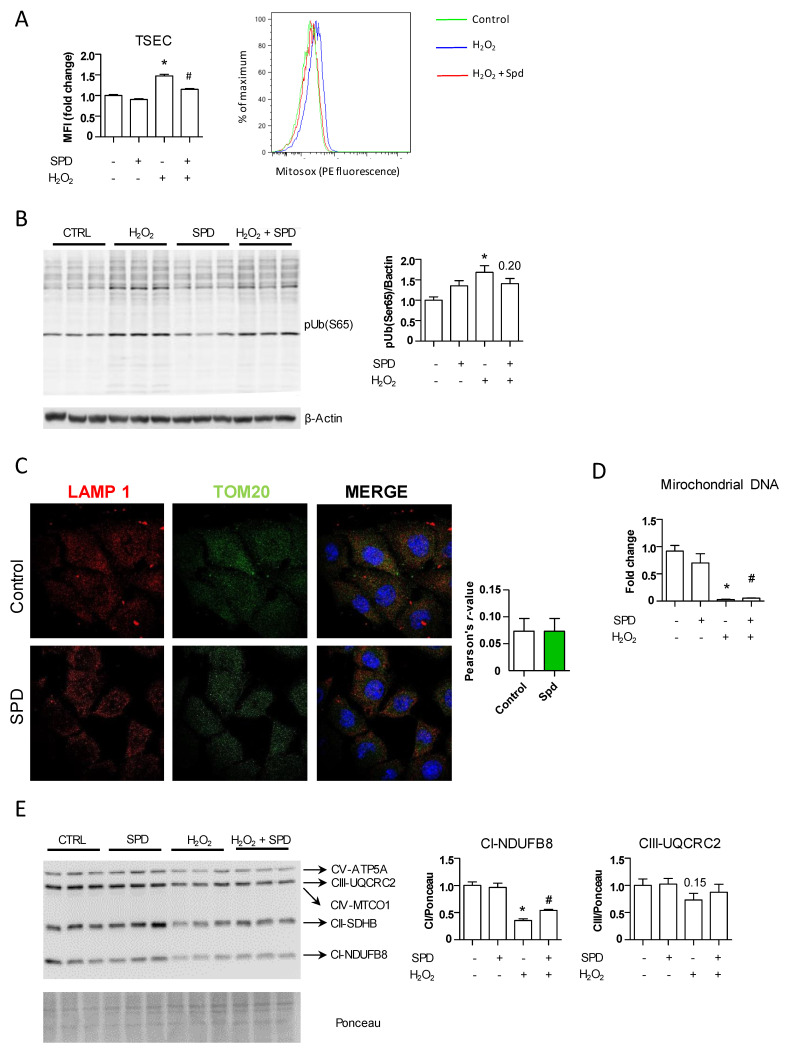
SPD protects mitochondria by reducing oxidative stress in vitro. (**A**) The TSEC were seeded in 6-well plates in triplicates. The next day, the cells were pretreated with SPD for 24 h. The medium was then changed, and H_2_O_2_ was added for another 24 h. The cells were then incubated with MitoSOXTM (2.5 μM for 10 min at 37 °C) before being trypsinized, washed, resuspended in DPBS + DAPI and analyzed by means of flow cytometry. One representative of the two independent experiments is shown. (**B**) The TSEC were treated as in (**A**). Twenty-four hours after the H_2_O_2_ addition, the cells were collected and analyzed by means of Western blotting against pUb(S65); β-actin was used as the loading control. One representative of the three independent experiments is shown. (**C**) Co-immunofluorescence of TOM20 and LAMP1 in the TSEC treated with SPD for 24 h. (**D**) Quantification of mitochondrial DNA by qRT-PCR from the TSEC treated as in (**A**). (**E**) Western blotting against the OXPHOS of the TSEC treated as in (**A**). Quantification of complexes I and III are shown. Ponceau staining was used as the loading control. One representative of the two independent experiments is shown. The results are the means ± SEM; * *p* < 0.05 compared to the correspondent control; # *p* < 0.05 compared to the H_2_O_2_ treatment (Student’s *t*-test).

**Figure 4 nutrients-13-03700-f004:**
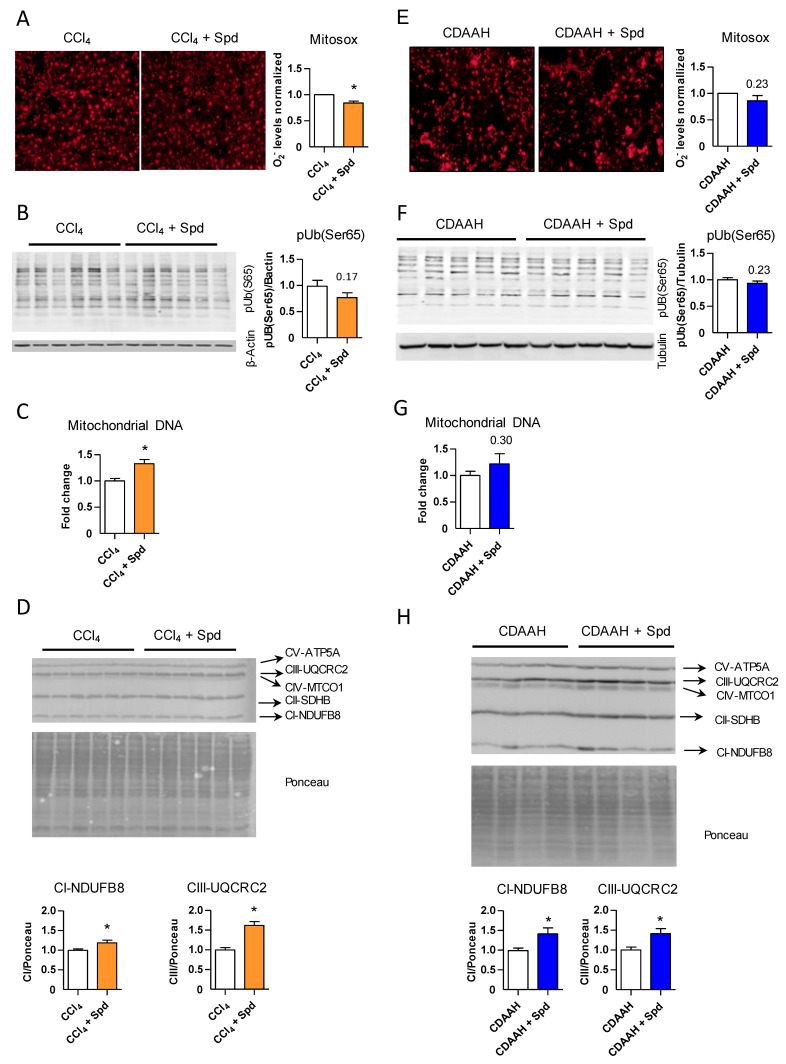
SPD alleviates oxidative stress and ameliorates mitochondrial stress signaling in vivo. (**A**,**E**) Superoxide quantification in the CCl_4_ and CDAAH mouse livers treated with or without SPD by MitoSOXTM. The mice were sacrificed in pairs (+/− SPD each time), the livers were excised and snap-frozen in dry ice. Ten μm cryosections, each containing a piece of the two livers, were obtained in less than an hour and rapidly incubated with a 5 μM MitoSOX^TM^ solution for 10 min in the dark. The slides were then covered with a coverslip and imaged within 10 min. Red fluorescence was measured from each liver and the SPD-treated liver value was relativized to its corresponding control. *N* = 4 for each model. (**B**–**F**) Western blot against pUb(S65) of livers from the CCl_4_ and CDAAH models; β-actin or tubulin were used as the loading controls. (**C**,**G**) Quantification of mitochondrial DNA by qRT-PCR in total liver tissue from the mice subjected to the CCl_4_ or CDAAH treatments with or without SPD supplementation. (**G**,**H**). Western blot against the OXPHOS of total liver tissue from the mice subjected to the CCl_4_ or CDAAH treatments with or without SPD supplementation. Ponceau staining was used as the loading control. Quantification of complexes I and III is shown. For every study, *n* = 7–10 mice per group unless indicated otherwise. The results are the means ± SEM; * *p* < 0.05 compared to the correspondent control (Student’s *t*-test).

## Data Availability

Not applicable.

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
