# Peer review of "Spermidine Supplementation Protects the Liver Endothelium from Liver Damage in Mice"

_nutrients, 2021, doi:10.3390/nu13113700_

Round 1

Reviewer 1 Report

Camprecios et al. examined the protective effects of dietary spermidine (SPD) against liver injury and its molecular mechanism. SPD supplementation in both CCl4-treated and CDAAH diet-fed mice improved the response to liver injury and attenuated fibrosis and HSC activation. In addition, SPD treatment also protects cells from hydrogen peroxide damage in TSEC and Sk-Hep1 cells. In vivo protective effect of SPD in the liver of CCl4-treated and CDAAH diet-fed mice was abolished in the Atg7endo mice, suggesting that autophagy enhancement is involved in the protective effect by SPD against liver injury. Furthermore, SPD protects mitochondria function against liver injury in normal mice but not in Atg7endo mice. Taken together, the authors suggest that supplementation of SPD ameliorates oxidative stress damage of liver endothelium and may be an attractive approach to limit the chronic liver disease process and halt fibrosis progression.

The reviewer thinks it is worth publication. The reviewer thinks that the primary data of the paper is Figures 4 and 5, and the involvement of autophagy is an associated finding. Therefore, Figures 3 and 6 should be included in supplemental data, and Figures 4 and 5 should be highlighted. Although autophagy is a required step for the improvement against liver damage, the first step of SPD function is to reduce ROS damage. Autophagy must be involved in the step of removing damaged molecules and organelle.

Additionally, the reviewer suggests some minor modifications as following described.

  1. If the first and second authors are the corresponding authors, add their addresses to the correspondence (Lines 4, 6, and 16).
  2. Correct superscript and subscript characters accurately in the manuscript. CCl4 should be CCl4, for example.
  3. Some abbreviations are used without a description (FBS, RIPA, and DAPI, for example). Please add the full descriptions.
  4. Please add the passage numbers. of TESC and SK-HEP1 cells used in the experiment
  5. Please add the catalog number of SPD the authors used and the description of how they made 1M SPD stock solution (Lines 85-87).
  6. Please add the reference of determination of liver transaminases (Line 90).
  7. Fig 1A, is data in the most right column necessary (+CQ)?

Reviewer 2 Report

In this study, the author investigated the effect of Spermidine supplementation on endothelium dysfunction under in vitro and in vivo conditions and found that SPD diet supplementation in early phases of liver disease protects the liver endothelium from oxidative stress. The authors further suggested SPD as a striking approach to modify the chronic liver disease course and halt fibrosis progression. The study is executed very well and results have been presented clearly.

 Minor comments

  1. The title of the manuscript is different in the main manuscript file and supplementary file. Please include the same title in both the files.
  2. Immunohistochemistry figures 1B, C and 2H are a bit blurry, replace them with better resolution pictures.
  3. Under in vivo studies using mice, please include the number of mice used in each set of experiments in the figure legend.    
